# A qualitative study examining stressors among Respiratory Therapists in Ontario amidst the COVID-19 pandemic

Marianne Saragosa[1]☯*, Farwa Goraya[1]☯, Behdin Nowrouzi-Kia[2,3,4]☯, Basem Gohar[1,4]☯

1 Department of Population Medicine, University of Guelph, Guelph, ON, Canada, 2 Department of Occupational Science & Occupational Therapy, University of Toronto Toronto, ON, Canada, 3 Temerty Faculty of Medicine, University of Toronto, Toronto, ON, Canada, 4 Centre for Research in Occupational Safety & Health, Sudbury, ON, Canada

☯ These authors contributed equally to this work.
* msaragos@uoguelph.ca

**Data Availability Statement:** All relevant data are within the paper and its Supporting Information.

**Funding:** This work was supported by the Public Services Health and Safety Association (https://

## Abstract

Health care systems were subjected to an unprecedented surge of critically ill patients with the coronavirus disease 2019 (COVID-19), which required management by Respiratory Therapists (RTs). Despite the high level of burnout reported in this health care professional group, we have limited knowledge about the lived experience of RTs during the pandemic. This study aims to examine the impact of COVID-19 on RTs in Ontario, Canada. We conducted a qualitative exploratory, descriptive study by conducting virtual semi-structured interviews and focus groups with RTs between March 2023 and June 2023. Two coders analyzed the data using thematic analysis. Twenty-seven RTs participated in the study, with the majority being female (n = 25), averaging 16.4 years of practice (range 4 to 36 years), primarily in acute care settings (n = 23). We identified four themes and lessons learned from the perspective of RTs: (1) *Working in the shadow and suffering in silence* reflecting varying perceptions of recognition; (2) *Flying blind amidst the buzz* reflecting the rapid pace of changing policies and practices as COVID-19 gained global attention; (3) *Putting out fires in the face of overflowing hospitals* reflecting increased workload and staffing issues; and (4) *Managing tensions*, *both external and internal* reflecting how RTs coped with distressing workplace situations and their mental well-being. Finally, lessons learned from the RTs include 1) Mobilizing early and consistently during an emergency, which addresses staff concerns; 2) Prioritizing and investing in the mental health and well-being of RTs; 3) Implementing strategies to retain experienced staff in healthcare; and 4) Involving RTs in leadership discussions. The COVID-19 stressors of RTs have illuminated the detrimental impact of the pandemic on this understudied health care profession. With this knowledge, targeted interventions can be developed to address RT recognition and staff retention and provide mental health support.

www.pshsa.ca/). The funders had no role in study
design, data collection and analysis, publication
decisions, or manuscript preparation.

**Competing interests:** The authors have declared
that no competing interests exist.

## Introduction

Respiratory Therapists (RTs) provide specialized respiratory care and education to patients
across their lifespan in various clinical areas [1]. In intensive care, RT's primary function is
facilitating airway management and mechanical ventilation [2]; however, RTs' training and
broad skill base enable them to provide other specialized procedures in primary care [3],
chronic disease management [4], and in specialized roles within the profession (i.e., cardiopul-
monary perfusionists, physician assistants, and anaesthesia assistants) [5]. As of 2023, there
were approximately 3,950 practicing RTs in Ontario [6], and now more than ever, the
increased need for pulmonary management to support the long-term effects of COVID-19 is
being recognized [7].

The COVID-19 pandemic, caused by severe acute respiratory syndrome coronavirus-2, has
led to unprecedented cases and deaths worldwide [8]. The early pandemic response consisted
of non-pharmacological interventions to limit the spread of the infection [9]. Similarly, hospi-
tals responded to extreme challenges—COVID-19 preparedness, managing and taking care of
healthcare workers, and logistical considerations [10]—with various mitigation strategies to
care for healthcare workers and maintain or expand hospitals' capacity to care for patients
[11]. Due to COVID's transmission, droplet/contact, and airborne transmission, hospital pre-
cautions targeted aerosol-generating procedures (i.e., endotracheal intubation, bronchoscopy
or airway suctioning, etc.) [12], in addition to managing visitors and operational controls (e.g.,
universal masking, monitoring systems, etc.) [13, 14]. Many procedures provided by RTs often
generate aerosolized airborne particles, leading to changes in their respiratory care practices
[15] and clinical management of COVID-19-positive patients (e.g., mechanical ventilation,
proning) [16]. The virus undoubtedly overwhelmed healthcare systems [17]. In a pooled Inten-
sive Care Unit (ICU)-admission and mortality rates were 21% and 28.3%, respectively [18].

Surges in critically ill patients [19], combined with staffing shortages and concerns about
inadequate access to personal protective equipment and ventilators [20, 21], have added to
stress levels among healthcare professionals. In particular, RTs' COVID-19 exposure and case-
load may make them susceptible to higher levels of psychological distress [22–24]. In a cross-
sectional study, high estimates of burnout were found in the surveyed RTs, regardless of the
presence of COVID-19 hotspots [23] and were higher than those of other clinicians [25]. Simi-
larly, investigators found in a Canadian RT sample clinically relevant symptoms of depression,
anxiety, and stress associated with functional impairment [26]. Other studies without COVID-
19 have demonstrated posttraumatic stress [27], moral distress [28], and depersonalization
[29] in staff RTs. With the "leadership" style being the most frequently identified and "pay" the
least identified driver for stress in RTs [24], it is imperative to understand COVID-19 stressors
and RT-generated strategies to address such factors moving forward from the pandemic. We
conducted semi-structured interviews and focus groups with practicing RTs in Ontario, Can-
ada, from March 2023 to June 2023, which, by then, 76% of the Canadian population had the
infection [30]. Therefore, this study provides an in-depth exploration of the perceived stressors
related to the experience of practicing RTs in Ontario, Canada, during COVID-19, focusing on
the lessons learned from the pandemic. The impact of COVID-19 is far-reaching, as skilled RTs
consider leaving their position [31] and suffering significant psychological distress [26]. Thus,
this study contributes to an essential gap in our understanding of how to better support the
needs of a workforce, which is invaluable to population health and during public health crises.

## Methods

This study used a qualitative descriptive design to understand the lived experience of RTs
working during the pandemic [32]. This study design was chosen due to its flexibility,

exploratory nature, and ability to investigate a phenomenon that is poorly understood by direct sources [32]. This study design has been used to develop or refine interventions in previous health care studies [32, 33]. All methods used for this study were reviewed and approved by the University of Guelph's Research Ethics Board (REB#22-03-001).

## Participants

Participating RTs were recruited from professional regulatory bodies and associations, snowball sampling or social media. Initially, participants were recruited through professional associations and regulatory bodies, which informed members about the authorized study materials via internal networks (i.e., listservs). Using snowball sampling, recruited participants shared the study with other RTs. Last, social media platforms, including LinkedIn and X (formerly known as Twitter), were used to share the recruitment flyer and invite additional participants. All participants were recruited in February 2023. For their participation, each participant received a $50 Amazon electronic gift card following their participation. To reduce fraudulent activities with online recruitment, participant identities were verified through public records and confirmation of their job roles pre-interview.

## Interviews

The interviews were conducted by one member of the research team virtually using videoconferencing software (Microsoft Teams Version in Office 365) between March 2023 and June 2023. The research team developed research questions based on experience in qualitative research, workplace mental health, pharmacy studies, and public health (S1 File). The interviews were semi-structured, starting with demographic questions, followed by open-ended questions regarding participants' experience working before and during the pandemic, the impact on professional and personal life, and lessons learned from the pandemic through an RT lens. As the study progressed, changes were made to the guiding questions to explore emerging themes. Before the interview, participants were informed they had the right to refuse to answer questions and/or withdraw from the study without any consequences. All participants provided written informed consent via email before starting their interviews. Ethics approval was obtained from the University of Guelph's research ethics board (REB#22-03-001, Approval date: 04 May 2022).

## Data analysis

Research team members (FG, MS) transcribed the audio recordings following the interviews. The collected text data was analyzed inductively using the six stages of thematic analysis, as proposed by Braun and Clarke [34]. More specifically, researchers conducted codebook thematic analysis [35]. Two coders (FG, MS) coded the transcribed text independently to ensure validity. First, the first five transcripts were coded using 'open codes' to find recurring and meaningful messages within the text (MS, FG). These open codes were discussed amongst the coders, who collected the codes and co-developed a codebook [35]. The remaining transcripts were divided between the two coders who used the codebook to continue the process. Consistent meetings occurred between the coders to discuss and refine codes. After coding all transcripts, the code names were readjusted and grouped by themes. The themes were constructed using the words of the participants from the text. The authors assumed data saturation when the interview information became repetitive and had no new information [36]. All data analyses were performed manually using Microsoft Word 365 on Mac.

**Table 1. Participant characteristics.**

| Characteristic | Finding |
|---|---|
| Age (Years, range, mean) | 27–57, 42 |
| Gender (Female n, %) | 25, 93% |
| Marital status (n, %) | |
| *Married* | 21, 78% |
| *Single* | 5, 18% |
| *Separated* | 1, 4% |
| Years in practice (range, mean) | 4–36, 17 |
| Years in current workplace (range, mean) | 3–34, 14 |
| Practice setting (n, %) | |
| *Hospital* | 25, 93% |
| *Community* | 2, 7% |

## Results

Of the twenty-seven participants, twenty-one participated in a one-on-one interview, and six participated in focus groups comprising three RTs each. The difference in the interview method was due to availability and preference. Most participants were identified as female (n = 25), practicing for an average of 16.4 years (range, 4–36 years), and in an acute care setting (n = 23). Table 1 presents the demographic characteristics of the participants.

The analytical process enabled the identification of four themes and lessons learned from the perspective of RTs. Themes consisted of (1) *Working in the shadows and suffering in silence* representing varying perceptions of recognition; (2) *Flying blind amidst the buzz* representing the rapid pace of changing policies and practices while the emergence of COVID-19 around the world started to garner attention a buzz; (3) *Putting out fires in the face of overflowing hospitals* representing the increased workload and staffing concerns; and (4) *Managing personal and professional tensions* representing how RTs dealt with distressing workplace situations and their mental health and well-being (see Table 2). Finally, lessons learned according to the RTs consisted of 1) *Mobilizing early and consistently during an emergency*, which addresses staff concerns; 2) *Prioritizing and investing in the mental health and well-being of RTs*; 3) *Doing more to retain staff to stop the drip of experienced staff leaving health care*; and 4) *Bringing RTs to the leadership table*.

### Theme 1: Working in the shadows and suffering in silence

Participants described their experience working as an RT as "working in the shadows" (P001) or identifying varying levels of their role and skills recognized by their colleagues, the patients, the organization and the public throughout the pandemic. Consequently, some felt the lack of recognition resulted in being under-resourced in staffing and pay. They also felt overlooked for handling the increased patient acuity and volume in hospitals when they did not qualify for the temporary pandemic pay noted in the following quote:

> "It would be said occasionally that RTs were vital, and then during that, the government released that they would be giving out stipend of pay, like a lump sum... We were, as an RT group, it felt terrible to work this hard and not be recognized…." (P008)

While some RTs expressed feeling frustrated, "...because we weren't always included in the equations, they weren't always considering that an RT was needed to be available to staff the beds or extra equipment for ventilators" (P014), only some RTs reported being sought by

**Table 2. Summary of findings.**

| No | Theme | Focused coding | Illustrative Quote |
|---|---|---|---|
| 1 | *Working in the shadows and suffering in silence* | •Going from invisible to being seen<br>•Feeling unrecognized<br>•RT role is behind the scenes | *"I think we work through the shadows, a lot of people like the doctors that dealt with us knew us. But a lot of people didn't really know what RT's were and, like, patients when you introduce themselves as a respiratory therapist, they'd be like, I'm not sure what that is. Everyone calls you a nurse."* (P001) |
| 2 | *Flying blind amidst the buzz* | •Sensing paranoia<br>•Hearing a lot of buzz<br>•Having doubts in new protocols | *"We didn't really know how to really handle it. Do we intubate them quickly? Do we leave them on these high amounts of oxygen? For everything else, we have these protocols where it's been tested and we know kind of exactly how to treat it. But with this it was, you were just kind of like going with itThere was some data coming out of, I think Italy, with the early waves that they had. And then we would see these people that we would intubate and get sick and worse so fast, that I remember questioning it around like that time, earlier on. Is this right? Like are we doing the right?"* (P008) |
| 3 | *Putting out fires in the face of overflowing hospitals* | •Increasing workload<br>•Intubations increasing<br>•Heavy demands | *"There was no patient care. There was absolutely no time for patient care. At all. It was literally just putting out fires. Cause you would have, like we went from an ICU with twelve beds to twenty-six. And every single person was ventilated."* (P002) |
| 4 | *Managing personal and professional tensions* | •Moral distressing situations<br>•Seeing younger patients very ill | *"It was it a year ago or two years ago, I was very stressed out about everything in life. I would say majority of it was because of work. I was having trouble with my sleep and anxiety level. Just like anxious all the time and I tried to reach out, because our hospital started offering those. What are they called? Like counseling or? Something like that, resources for staff. And I did sign up, but I didn't really go through with it because it wasn't talking to a real person. It was more set up in a way that you have to go through learnings and modules yourself and then checking in with the, was a social worker or somebody."* (P007) |

hospital leadership to contribute their knowledge about respiratory supplies and procedures. Despite RTs' in-demand expertise, they continued to notice the absence of their recognition in the media. This highlights that lack of recognition reached a 'peak' for some RTs who felt that despite their crucial role during the pandemic, their efforts were dismissed similar to the pre-pandemic era.

> *"I think what I found different is always in the media. Well, nurses and doctors and nurses and doctors. At one point, I'm maybe because I was exhausted and, like, we're there too. We're there too, and we're never mentioned. But I thought, okay, physios are there too, and they're never mentioned either, but I just felt like at this time I was just getting, I think because you're so tired and maybe sad that the anger comes out so much more readily then. You're just frustrated."* (P005)

Conversely, some RTs observed shifting public awareness of the profession from a positive to a more pessimistic viewpoint. One RT who works in the community related this shift to polarizing public opinions of the pandemic and vaccine. For example, two RT participants expressed the following reflections,

> *"During the pandemic, I feel like RTs were held as heroes. But we weren't recognized, I think enough. But we were looked at, people knew what respiratory therapists were. . .For the first time in my career, walking around, people knew we existed. . .As time has passed, I've seen comments about respiratory therapists that are completely shocking. I've seen it, where people are talking about patients on a ventilator. Ventilators kill people. You people have killed people on ventilators. It's things that are just completely inaccurate and off base."* (P010)

> *"There was this massive surge of community support in the first wave. People honking their car horns outside of the hospital, like just trying to show support, and then, as it got gruelling and the teams could have used that, you didn't see it anymore. Instead, you saw anti-vaxxers protesting outside your hospital."* (P013)

## Theme 2: Flying blind amidst the buzz

As the pandemic emerged in March 2020, RTs remarked on "paranoia" and a "buzz around the hospital" (P002) as crisis management plans unfolded. However, many participants noted the global response to the virus and attempted to draw parallels to their own situation, illustrated in the following quote:

"*I was consuming so much news and so much from Twitter, like the peak had started subsiding in Italy when we were getting many of our patients. And then New York was blowing up, and I was like, 'guys, I fear that we're not going to be Italy, we're going to be New York.'*" (P003)

RTs also described the once busy hospital scene as "eerie" (P002), or another RT called it "the calm before the storm" in the following quote, "*And then I remember very clearly that it felt very quiet, for about like two weeks in the hospital. We had fewer patients in the ICU than I had remembered in the previous couple of years that we had worked. There was also a lot of anticipation and fear among the staff*" (P007).

Many participants' primary concern was the potential shortage of personal protective equipment (PPE). Collective fears of the virus caused PPE to be taken by patients, visitors, and staff, which resulted in supplies being locked away. In response to curtailing declining PPE supplies, RTs also mentioned the extended use of respirators, often placing them in bags with other labelled supplies:

"*I know there were constantly changing things for the PPE. And at one point, they were telling us to reuse our masks, which I thought was ridiculous. They gave us paper bags, and they're like, put your mask in this paper bag. And I think, okay, you put it in the paper bag. You now contaminate the inside of that paper bag, and it's also touching the inside of your mask.*" (P007)

Another commonly reported feeling was the sense of RTs it was a "fly by the seat of your pants kind of situation" (P017). The lack of information about the virus, combined with the frequently changing policies and practices, caused significant stress and anxiety in the participants. Many remarked on the changing practices concerning infection control, caring for COVID patients, and handling code blues, "*It was pretty hectic, and we got a lot of information, but the information was changing all the time*" (P023). Some RTs identified inconsistent communication or conflicting guidelines as particularly taxing:

"*But what was unnerving was not having the information come to you, like not knowing what the what to do, and having the information changed so frequently, that, you know, you've come and being told by different people like, no, no, no, you're doing it wrong, like this, you know, and being yet like, just, it was just so willy nilly, haphazard. And that was what was frustrating and scary. It's like, what are we doing? Let's figure it out and stick to something. I mean, one day, it's non-N95s, or whatever the next minute it is, and, you know, so that was that, that I found was tricky.*" (P017)

With new research emerging, RTs changed their own practice in response; however, this also caused some to reflect on earlier handling of COVID-positive patients, "*All the research for the first wave was intubating early, which is honestly traumatizing to think of, like as soon as I hit fifty percent oxygen, intubate, intubate, intubate. And then, months later, research came out*

*that showed a significantly higher mortality for intubated patients, so that was awful to think back*" (P019).

## Theme 3: Putting out fires in the face of overflowing hospitals

The hospital surge among critically ill patients with COVID-19 resulted in increased workload, compromised care, and concerns over the workforce, as well as PPE and equipment shortages. For the former, RTs described a constant busyness with taking on more "time-consuming" practices like proning and head turns and the use of high-flow oxygen,

> "*There was no patient care. There was no time for patient care. At all. It was just putting out fires. We went from an ICU with twelve beds to twenty-six. And every single person was ventilated.*" (P002)

They also reported taking more time donning and doffing PPE when treating patients, "*Like if there's a code blue called, you have to put on your N95, gown, and gloves. Like, people were putting on bonnets and stuff like that. So, I feel like we weren't as quick to treat, which makes me sad*" (P022). Many RTs also recounted stories of COVID-19-positive patients experiencing a rapid decline in otherwise healthy or younger populations or the virus infecting and killing entire families.

> "*But when you're looking at a forty-five-year-old otherwise healthy person with a family of three, and they've got the iPad running in the corner of the room, and these kids are bouncing around and looking at their dad who's prone and like, all the therapies going. You're like, oh my gosh, we have to. There has to be a good outcome for this.*" (P006)

Other RTs noted that during wave one, they received patient transfers from community hospitals, and these patients required a higher level of ventilation, or their COVID patient census "*skyrocketed*" (P007) during waves two and onward, which is illustrated in the following quote:

> "*So proning became a very routine practice after COVID. The guideline is to turn the head every two hours, but we just have no capacity, so that was changed to every four hours. But even with that, you're constantly in the unit without having a break. You can't remove your mask because you're scared of being infected. You can't take water, you can't eat. And it's just like everybody's asking you, and there was a lot of stress from that, too, because everything is an emergency.*" (P007)

Working shifts like this caused anxiety in RTs who felt overwhelmed because of the patient volume and inadequate staffing, "*Like you didn't even have a chance to, like, catch your breath. No one available to help you because they're all busy too*" (P018). According to one RT, when they did return to work, the patient turnover was so quick, "*I remember I'd intubate like two or three patients sometimes in a shift, and then I'd come back a day or two late, and they'd be gone. And no one would even know where they went. I'm like, did they die? Did they get transferred?*" (P004). In the summers since COVID-19, RSV and influenza on top of COVID replaced lower patient volumes. Working through the third to fifth waves left some RTs with little work-life balance.

Several RTs mentioned a lack of leadership presence either because they perceived leadership was "*too busy*" (P014), the RT manager "*got pulled to another hospital at the worst time ever*" (P019), or they refused to listen to advice about ordering supplies, "*So she kept making*

*me feel like I was crazy*" (P021). In response, the RTs assisted and supported each other, "*So the hospital or the higher-ups didn't see any problems because we would help each other do team coverage, and we would still cover everyone safely*" (P014).

## Theme 4: Managing personal and professional tensions

According to one RT, the combination of the patient volume and acuity and being unsure what to do "*made it all very explosive*" (P010) and caused "*. . .a lot of friction between staff members. . .lots of conflict*" (P002) and moral distress. Heightened emotions spurred tensions and contributed to some RTs managing personal and professional stressors. For example, several participants remarked on internal tensions within the RT department or, more broadly, the hospital setting about receiving the COVID-19 vaccine, vigilance about infection control practices, and the pandemic pay. One RT observed internal departmental fractures,

"*You could see the beginning of the disconnect like people maintaining restrictions and guidelines. Some people were not. There were people, there was a division in our department because individuals caused outbreaks. And then there was a ripple effect from the outbreak of the potential of impacting our loved ones who are immunocompromised or elderly.*" (P006)

Another source of tension was the mandated COVID-19 vaccine within health care settings. For some RTs, they tried to rationalize deciding against the vaccine,

"*. . . there was a bunch of people that decided they didn't want to get vaccinated, and when they decided to make vaccines mandatory, that created a huge divide, and then people developed hate, dislike or distaste toward anyone who did not have a vaccine. And then, to top that off, these people lost their jobs.*" (P017)

Frustrations were also felt because of the public response to masking and vaccine mandates. RTs commented on public protests in relation to their own challenges with caring for COVID patients, "*And I remember the patients coming in, and when they were anti-maskers and anti-vaxxers, that's when it got frustrating. We had two brothers who both died because they refused to wear masks. And then they went out. Okay, if you don't want to wear a mask, that's fine, but stay home. You know, like, they'd go in public places*" (P004).

Feelings of moral distress were also reported by many who expressed concerns about not being able to use certain respiratory therapies because of restrictions or applying therapies on futile cases, "*So mentally, we were all like, this is futile. Like, why are we doing this? Why do we keep providing this? At this point, it's a death sentence for these people. Like this is just even more torture for them, because they're already suffering, and now we are prolonging their life but not saving their life*" (P020).

Personally, RTs also struggled with worries about becoming infected and transmitting the virus to their family. While some RTs had concerns about their children's mental health, balancing online learning and work, others expressed gratitude when their partner was able to work from home and care for their children, "*So by him working at home helped me a lot because my daughter was at home, so I didn't have to worry about taking care of my daughter at that time*" (P023). However, many participants disclosed significant impacts of COVID-19 on their mental and physical health, including having panic attacks, dysregulation, and reporting unhealthy coping mechanisms like drinking alcohol. One RT described the trauma they experienced,

*"Then, once you're done with that patient, you go right to the next person. There's no break. There's no point where you're processing it. What you're doing is you're just tucking it to the back of your head. And I think during COVID, because the amount and the acuity had gone so high, the demand was so high, not only were you not processing it, but you were tucking it all away even more. At a faster rate. And eventually, it caught up with you; in the back of your head, it would."* (P010)

## Lessons learned

The RT participants identified several lessons learned from practicing during the pandemic (see Table 3).

**Mobilizing early and consistently during an emergency, which addresses staff concerns.** The lesson on pandemic preparedness was overwhelmingly decisive. Many RTs acknowledged that undermining the virus initially as "like the flu" and not taking it seriously was a lesson learned, as was having appropriate staffing, supplies and PPE in place. For example, one RT said, "*So good practices that came out of it was they identified the need for equipment early, and they had access to it, so it's almost like a foreseeability piece. Like our organization now, after COVID, I feel it has a more attuned radar to situations that warrant increased staffing, equipment, and revision of guidelines and practices*" (P006). Another change in practice was that PPE and infection control practices would continue to be applied. "*I find with my other hospitals where the culture wasn't so strict about wearing an N95, I used to see people intubating without them. Now they always wear them. So that was probably one of the bigger things that changed*" (P004).

**Prioritizing and investing in the mental health and well-being of RTs.** Another significant insight gained from the pandemic was prioritizing and investing in mental health resources for RTs. Boosting wellness services was noted to be in the control of organizations and a means of recognizing the existence of workplace stress. However, it was also observed that wellness budgets are also first to be cut during times of austerity. Similarly, part-time staff may not have the same access to such services because of their benefit plans.

**Table 3. Summary of lessons learned.**

| No | Lesson learned | Focused coding | Illustrative Quote |
|---|---|---|---|
| 1 | *Mobilizing early and consistently during an emergency, which addresses staff concerns* | •Downplaying the virus<br>•Learning to monitor and evaluate staffing<br>•Being more diligent with PPE | *"So, the healthcare system really can just, be better prepared for a surge of that situation. So much equipment was on back order, was having to be bought and at that point, where if we had stuff on storage ready to go, it would have been a lot smoother that way."* (P008) |
| 2 | *Prioritizing and investing in the mental health and well-being of RTs* | •Hospital starting to offer more mental health resources for staff<br>•Being unable to get time off | *"Like you have to put a price on what your mental health is, right? And I'm thankful I've been doing it for as long as I have cuz it's helped me. But there's people that don't have that. Like the part timers didn't have any access to benefits, right? So, they did extend it to the part time staff and the casual. Everyone has fifteen-hundred dollars for mental health benefits, but it's only good for the end of the year and they're not extending it past that. And it's like, that's going to do nothing, but OK."* (P026) |
| 3 | *Doing more to retrain staff to stop the drip of experienced staff leaving health care* | •Needing to do more to retain staff<br>•Creating more ful-time lines<br>•Providing incentives to those who are training new staff | *"But it's like, what about the people that are here and have still been here and are the ones training all these people. 'Cause there's no extra money for training new staff, and that's extra work, right? And then they just leave so then you have to do it again. But it's like, you're having to do even more work because you have your normal assignment, plus trying to teach someone too, right?"* (P002) |
| 4 | *Bringing RTs to the leadership table* | •Recognizing RTs or else they stop caring<br>•Being embedded in those leadership conversations | *"I do think that the government does need to recognize that we all work better together. It's not just one. We're multidisciplinary and we know what we're doing. Like, they need to rely on our skills, our abilities."* (P010) |

**Doing more to retrain staff to stop the drip of experienced staff leaving health care.**
Many RTs commented that more needs to be done to retrain staff to remain in critical care
and recruit new staff. Suggestions ranged from recognizing RTs through pay increases,
acknowledging their efforts and offering more full-time employment options. Several RCTs
identified commuting to urban hospitals as a barrier with expensive parking fees.

"*The healthcare leaders and policymakers keep talking about getting more people into health
care by expediting internationally trained staff. It's all about recruitment. They never think
about retention. They never think about why people are leaving. They always say, oh, there's
not enough people working in healthcare. But it's because people are leaving. This work envi-
ronment isn't that great.*" (P007)

**Bringing RTs to the leadership table.**   Several RTs mentioned that RTs need to be at the
"big tables early" (P005). In effect, their voices need to be heard primarily for respiratory-
related issues. In the future, RTs want to be brought into conversations, "*Speak to us, speak to
the College of Respiratory Therapist. We have a lot to offer. We have a lot of good background,
good advice. I mean yeah, I think that you know, working together, rely on us*" (P010). They also
want more RT support in hospitals and the community to help manage chronic diseases.

## Discussion

This qualitative study investigating stressors during the COVID-19 pandemic provides an
insightful understanding of perceptions and experiences through the lens of RTs and several
lessons learned. While there is emerging evidence of RT burnout, stress and compassion
fatigue in response to COVID-19 [24, 37], more in-depth qualitative data must be collected
from larger RT samples [38, 39]. Severe acute respiratory syndrome coronavirus-2 causes
respiratory failure requiring oxygen therapy and/or mechanical ventilation, in which RTs are
trained to deploy; however, early experiences and predictive modelling during the pandemic
suggested the need for RTs to exceed the current supply [40]. This supports the urgent need to
understand the drivers of stress and develop targeted supportive measures after COVID-19.

COVID-19 research on mental health in healthcare providers has focused mainly on larger
healthcare professional practice groups, including physicians [41, 42] and nurses [43], with
less representation from allied health [44–46], including respiratory therapists [24]. Thus,
interventions designed and implemented to address mental health problems consider the
needs of frontline nurses and doctors [47, 48], potentially causing a misalignment between
what RTs want and need and the resources available to them. Our study, therefore, provided
an in-depth exploration of the perceived stressors related to the experience of practicing RTs
in Ontario, Canada, during COVID-19, focusing on lessons learned from the pandemic that
could help drive mitigation strategies to avoid burnout [22].

Our findings underscore the perception held by many RTs about the ongoing lack of recog-
nition and acknowledgment of their role organizationally, as well as by the public and the gov-
ernment. As a result, RT participants described resource issues as inadequate RT staffing
levels, unmanageable workload, and thankless gestures, such as initial exclusion from ministry
pandemic pay. Many concerns related to RT burnout, notably staffing, workload, compensa-
tion, and lack of respect, were found in an American RT sample [24]. The lack of recognition
and appreciation of the knowledge, skill, and resource difficulties of RTs has been cited in ear-
lier research [49]; however, it seems to have been exacerbated by the pandemic [50]. Approxi-
mately 79% of RTs surveys in 26 American hospitals experienced burnout Field [21], and a

large proportion experienced high rates of depression and minimal professional fulfillment [25]. Efforts to address burnout and the related negative impacts of compassionate fatigue and lack of recognition require collective strategies at the organizational level [22]. Consistently measuring staff well-being and burnout is a suggested starting point for organizations to identify precipitating factors [51] and ways to create healthy work environments [52].

The rapid change in COVID-19-related policies and practices during the initial period of the pandemic caused anxiety and stress among RTs. However, more than the rapidness of change was the inconsistent communication of changes and, at times, conflicting messaging. Many of these experiences have been shared by other healthcare professionals worldwide who described COVID-19 guidelines as frequently changing, lacking transparency, and at odds with healthcare professionals' values and pre-pandemic standards of care [53]. Our findings are consistent with organizational barriers to healthcare providers' adherence to COVID-19 protocols, which include unclear, inconsistent and changing protocols to which hospital providers are not adequately informed and oriented [54]. This is particularly troublesome given COVID-19's quick progression to respiratory failure and death [18] and the availability of emerging evidence to avoid specific therapies with unfavourable outcomes [55]. This finding provides an opportunity to improve pandemic preparedness by creating standardized bi-directional communication plans to deliver pertinent information and feedback to and receive from clinicians and system leaders [56].

As the pandemic started and continued, RTs experienced significant growth in workload alongside staff concerns and, accordingly, high mortality rates and morally distressing clinical situations. This is not unlike other healthcare professions in different jurisdictions that reported more tasks (i.e., donning and doffing PPE) and mental, physical and time pressures compared to non-COVID-19 patient exposure [57]. The increase in the volume of workload and insufficient RT staffing also impacted a sense of feeling overwhelmed and moral distress. Post-pandemic workforce data indicates many RTs have left their jobs, and 30% are considering leaving [58]. A prior survey by Strickland, Roberts (24) confirms this finding by finding high workload assignments negatively impacted the quality of care RTs provided during the pandemic. Elevated moral distress was also found in a Canadian sample of RTs (N = 213), with one in four reporting considering leaving their job because of morally distressing events [31]. The significance of moral distress on retention cannot be understated [59], yet evidence-based strategies to mitigate moral distress in RTs are absent [60]. The *Moral Stress Amongst Healthcare Workers During COVID-19*: *A Guide to Moral Injury* [61] is a tool for inspiring systematic efforts to prevent and intervene to avoid and reduce moral distress.

Broader organizational and system factors may also be critical in RTs' perceived organizational support and job commitment [62]. The lessons learned identified by RT participants in our study highlight the importance of both pandemic preparedness and retention efforts, as well as supporting well-being and valuing the expertise of RTs at a leadership level. Current policy briefings and career support programs reflect better support for RTs to retain them in the workforce. Policy recommendations by the Canadian Society of Respiratory Therapists outline six suggestions across three areas of need, including establishing a national mental health strategy for RTs and all healthcare providers [58]. Important practice initiatives have also been put forward in response to the RT workforce crisis. The Functional Residual Capacity framework creates a structured approach for leaders to consider how to support RTs in all clinical settings while New Graduate Transition Supports program in British Columbia is also demonstrating promising results [63].

Importantly, perceived ineffective leadership has been recognized as a common factor driving burnout among RTs [37]. Therefore, recognizing the importance of RTs beyond the current pandemic as a valuable role could be an important organizational strategy for retaining

RTs [25], especially since many of our participants and other research support RTs experienced severe work-life disruptions [38]. Notably, RTs interviewed strongly desired to be included in leadership discussions and decision-making processes. Creating capacity for more formal leadership positions in RT departments or hospital systems may positively contribute to the work environment, reduce the risk of burnout, and generally inform policies and practices. There is limited data to support the RT leadership practice [37], which strongly signals a future study area.

This study has several strengths and limitations. Our large sample size of RTs is a strength to profoundly understanding the lived experience of Canadian RTs during COVID. However, we acknowledge that we may have introduced sampling bias using a convenience sample of RTs. Therefore, findings may present an extreme view of the topic. However, to mitigate such biases, we attempted to recruit participants with diverse characteristics, such as years of practice and roles held as an RT in addition to the frontline, such as management. Lastly, only RTs practicing in Ontario, Canada, were included, and the experiences of RTs in other parts of the country and the world would likely be different.

## Conclusions

The RTs in this qualitative study described a series of stressors related to practicing during the pandemic. The stressors included a lack of recognition for their role in handling COVID-19, changing practices and policies without evidence about the virus, increased workload, morally distressing events and personal anxieties. Despite their poor experiences, RTs identified several lessons that could be applied as we move forward, focusing on crisis preparedness, prioritizing healthcare providers' mental health and well-being, retention strategies and leadership opportunities for frontline RTs.

## Supporting information

**S1 File. RT study interview guide.**
(DOCX)

## Acknowledgments

We acknowledge the contribution made to the review of this paper and substantive support for recruitment from Carolyn McCoy, a staff member of the Canadian Society of Respiratory Therapists. We also acknowledge the valuable contribution and stories of the respiratory therapist participants.

## Author Contributions

**Conceptualization:** Basem Gohar.

**Data curation:** Marianne Saragosa, Basem Gohar.

**Formal analysis:** Marianne Saragosa, Farwa Goraya, Basem Gohar.

**Investigation:** Basem Gohar.

**Methodology:** Marianne Saragosa.

**Project administration:** Marianne Saragosa.

**Writing – original draft:** Marianne Saragosa, Farwa Goraya.

**Writing – review & editing:** Marianne Saragosa, Farwa Goraya, Behdin Nowrouzi-Kia, Basem Gohar.

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
