## [Decision Letter · Decision Letter 0]

25 Apr 2024

PONE-D-24-05435Breathing Under Pressure: Examining Stressors Among Respiratory Therapists in Ontario Amidst the COVID-19 PandemicPLOS ONE

Dear Dr. Saragosa,

Thank you for submitting your manuscript to PLOS ONE. After careful consideration, we feel that it has merit but does not fully meet PLOS ONE’s publication criteria as it currently stands. Therefore, we invite you to submit a revised version of the manuscript that addresses the points raised during the review process.

We look forward to receiving your revised manuscript.

Kind regards,

Petra Czarniak, PhD

Academic Editor

PLOS ONE

Journal Requirements:

Reviewers' comments:

Reviewer's Responses to Questions

**Comments to the Author**

1. Is the manuscript technically sound, and do the data support the conclusions?

Reviewer #1: Yes

Reviewer #2: Yes

2. Has the statistical analysis been performed appropriately and rigorously? 

Reviewer #1: N/A

Reviewer #2: N/A

3. Have the authors made all data underlying the findings in their manuscript fully available?

Reviewer #1: Yes

Reviewer #2: Yes

4. Is the manuscript presented in an intelligible fashion and written in standard English?

Reviewer #1: Yes

Reviewer #2: Yes

5. Review Comments to the Author

Reviewer #1: Saragosa et al performed a qualitative study evaluating stressors among RTs working in Canada. This is an excellent study and well-written. I have a few questions and comments below to help improve the quality of the work. Importantly, there are some missing citations from the RT literature on wellness. Lastly, are any of the authors RTs? If not, I would suggest asking an RT for further insight to sharpen the discussion with more specifics on what can be done to help the mental wellness of RTs.

Title: Please remove Breathing Under Pressure, while RTs help people breath, the subjects of this study are the RTs themselves.

Are any of the authors RTs?

Abstract

Background can be written more clearly. The pandemic caused massive surges in critically ill patients requiring mechanical ventilation, which overloaded RT departments and caused very high workloads.

Manuscript

Page 4, line 88 requires citation and specific examples of psychological distress. Would suggest Miller et al doi:10.4187/respcare.09283 and Strickland et al doi: 10.4187/respcare.10144 or a review article from the same group Miller et al doi: 10.4187/respcare.10632.

Reference 22 was not a study of burnout but rather of what supportive resources were available. Would change reference to Miller et al doi:10.4187/respcare.09283.

Citations are needed for pay and leadership being drivers of burnout.

Methods

Very nicely written.

Results

First sentence starts with a lower case of, is there part of the sentence missing?

What do you mean by 27 participants who led interviews? Does this mean participated?

Do you think the interviewees were representative of the total population of RTs? What role did they have in their hospital? The views of front-line clinicians in the ICU will likely be different than leaders or staff less exposed to COVID-19.

Why was this categorized as working in the shadows and suffering in silence instead of the simpler definition like (lack of appreciation or acknowledgement)?

The others themes are similar, it seems the authors are trying to be clever instead of simply describing what is happening.

The rest of this section is very nicely done. I particularly find the quotations quite insightful and valuable.

The discussion about retention over recruitment is spot on and a sore spot where I work where all the resources go to recruitment.

I would suggest adding some tables or figures to help more clearly present the results.

Discussion

While additional qualitative research would be helpful, available quantitative surveys have identified a number of factors associated with burnout (which can be used as a marker of stress). I would suggest mentioning them, drawing on the review article from Miller et al cited above.

I would reframe a lot of this discussion to address post-pandemic systemic issues such as staffing, emotional support, and other factors respiratory care departments and hospitals can do to better support RTs.

In the limitations section, your sample is actually quite small and you need to present more data (or acknowledge it is non-representative) about the respondents.

Reviewer #2: Well developed study. It's good to see more qualitative work in the field.

Methodology:

Were the interviews conducted by one team member or more than one? How were the interviewers trained to reduce variability among the individual interviews/focus groups?

I assume that you used traditional text coding (versus electronic coding software) but that isn't explicitly stated.

Nice incorporation of direct quotes to demonstrate relevancy and give voice to the participants.

Would like to see the actual interview questions included in the manuscript as a supplementary document.

Minor corrections:

Line 118: I think you mean "formerly known as Twitter" (not formally)

Line 150: Is there a word missing at the beginning of the paragraph or is the "O" in "of" supposed to be capitalized?

Line 306 - not sure why anxiety is in quotations.

Line 312-313 - looks like sentence fragment

6. PLOS authors have the option to publish the peer review history of their article (what does this mean?). If published, this will include your full peer review and any attached files.

Reviewer #1: No

Reviewer #2: **Yes: **Shawna Strickland, PhD, RRT, FAARC

---

## [Author Response · Author response to Decision Letter 0]

11 Jul 2024

May 26, 2024

Dear Reviewers,

We appreciate you and the reviewers for reviewing our paper (Breathing Under Pressure: Examining Stressors Among Respiratory Therapists in Ontario Amidst the COVID-19 Pandemic) and providing valuable comments. The insightful comments have led to improvements in the current version. The authors have carefully considered the comments and worked to address each in a point-by-point table below. 

All the modifications are below and in the manuscript with track changes. The authors welcome further constructive comments on the clarity of our arguments, the relevance of our references, and any other aspect that could enhance the quality of our paper.

---

## [Decision Letter · Decision Letter 1]

8 Oct 2024

A Qualitative Study Examining Stressors Among Respiratory Therapists in Ontario Amidst the COVID-19 Pandemic

PONE-D-24-05435R1

Dear Dr. Saragosa,

We’re pleased to inform you that your manuscript has been judged scientifically suitable for publication and will be formally accepted for publication once it meets all outstanding technical requirements.

Kind regards,

Emily Lund

Academic Editor

PLOS ONE

Additional Editor Comments (optional):

Reviewers' comments:

Reviewer's Responses to Questions

**Comments to the Author**

1. If the authors have adequately addressed your comments raised in a previous round of review and you feel that this manuscript is now acceptable for publication, you may indicate that here to bypass the “Comments to the Author” section, enter your conflict of interest statement in the “Confidential to Editor” section, and submit your "Accept" recommendation.

Reviewer #1: All comments have been addressed

Reviewer #2: All comments have been addressed

2. Is the manuscript technically sound, and do the data support the conclusions?

Reviewer #1: Yes

Reviewer #2: Yes

3. Has the statistical analysis been performed appropriately and rigorously? 

Reviewer #1: N/A

Reviewer #2: Yes

4. Have the authors made all data underlying the findings in their manuscript fully available?

Reviewer #1: Yes

Reviewer #2: Yes

5. Is the manuscript presented in an intelligible fashion and written in standard English?

Reviewer #1: Yes

Reviewer #2: Yes

6. Review Comments to the Author

Reviewer #1: The authors have satisfied my concerns.

Reviewer #2: Thank you for addressing my concerns. I do think that the methodology could be further enhanced but the necessary information is provided.

7. PLOS authors have the option to publish the peer review history of their article (what does this mean?). If published, this will include your full peer review and any attached files.

Reviewer #1: **Yes: **Andrew G. Miller

Reviewer #2: No

---

## [Editor Report · Acceptance letter]

30 Oct 2024

PONE-D-24-05435R1 

PLOS ONE

Dear Dr. Saragosa, 

I'm pleased to inform you that your manuscript has been deemed suitable for publication in PLOS ONE. Congratulations! Your manuscript is now being handed over to our production team.

Kind regards, 

on behalf of

Dr. Emily Lund 

Academic Editor

PLOS ONE